# Leveraging prior knowledge to infer gene regulatory networks from single-cell RNA-sequencing data

Marco Stock[1,2,3,4], Corinna Losert [2,5,7], Matteo Zambon [1,2,3,7], Niclas Popp[1,2,3], Gabriele Lubatti[1,2,3], Eva Hörmanseder[1], Matthias Heinig[2,5,6] & Antonio Scialdone [1,2,3✉]

## Abstract

**Many studies have used single-cell RNA sequencing (scRNA-seq) to infer gene regulatory networks (GRNs), which are crucial for understanding complex cellular regulation. However, the inherent noise and sparsity of scRNA-seq data present significant challenges to accurate GRN inference. This review explores one promising approach that has been proposed to address these challenges: integrating prior knowledge into the inference process to enhance the reliability of the inferred networks. We categorize common types of prior knowledge, such as experimental data and curated databases, and discuss methods for representing priors, particularly through graph structures. In addition, we classify recent GRN inference algorithms based on their ability to incorporate these priors and assess their performance in different contexts. Finally, we propose a standardized benchmarking framework to evaluate algorithms more fairly, ensuring biologically meaningful comparisons. This review provides guidance for researchers selecting GRN inference methods and offers insights for developers looking to improve current approaches and foster innovation in the field.**

**Keywords** Gene Regulatory Network Inference; Prior Knowledge; Single-cell Transcriptomics; Single-cell Multiomics; Graph Learning
**Subject Categories** Chromatin, Transcription & Genomics; Computational Biology

## Introduction

The increasing availability of large-scale single-cell RNA-sequencing (scRNA-seq) datasets (Svensson et al, 2020) has driven the development of numerous computational methods to infer gene regulatory networks (GRNs). scRNA-seq data offer unique insights into cell-to-cell variability that are obscured in bulk RNA-seq datasets, making them particularly valuable for understanding the intricate regulatory mechanisms underlying biological processes.

The identification of GRNs provides critical insights into the causal relationships that govern gene interactions, revealing which genes are pivotal in specific contexts. However, constructing GRNs from scRNA-seq data is challenging for several reasons, including the presence of biological (e.g., cell-cycle-related (Buettner et al, 2015; Liu et al, 2021)) and technical confounding factors (e.g., high sparsity and high level of intrinsic noise and dropouts (Qiu, 2020)), as well as GRN properties, like the presence of feedback loops (Atanackovic et al, 2023). Many algorithms have been proposed to tackle this complex problem (Hawe et al, 2019; Pratapa et al, 2020). Nevertheless, the consensus of recently published benchmarking studies is that the performance of GRN inference algorithms is still limited (Kang et al, 2021; McCalla et al, 2023; Pratapa et al, 2020; Stock et al, 2024). (Pratapa et al, 2020) find highly variable and overall poor performance of the algorithms across several datasets. (Kang et al, 2021) highlight poor reproducibility of inferred GRNs, even from independent datasets collected under the same biological condition. (McCalla et al, 2023) demonstrate that advanced approaches cannot consistently outperform simple linear correlation in their analysis. (Stock et al, 2024) show that the available algorithms include topological biases in their inferred GRNs.

A promising strategy to improve this is the incorporation of prior knowledge into the inference process (Liu, 2018; McCalla et al, 2023). For instance, prior knowledge could involve known regulatory interactions between specific gene pairs, or the use of multi-omic data, which can provide information on chromatin accessibility, maps of DNA physical contacts, etc. Integrating such information can enhance GRN inference, e.g., by constraining the solution space or by providing labeled examples that the algorithms can use for learning.

Numerous sources of prior knowledge are available, as well as various approaches for incorporating them into the inference process. This diversity has led to the rapid development of a wide range of algorithms, each offering distinct combinations of strategies. As a result, it has become increasingly challenging to keep track of the available algorithms, the types of prior knowledge they utilize, their computational methodologies, and their performance across different datasets. For users, this complexity poses challenges in selecting the most suitable algorithm for their needs, while developers face obstacles in identifying weaknesses and areas for improvement. These issues are further compounded by the lack

[1]Helmholtz Center Munich Institute of Epigenetics und Stem Cells, Munich, Germany. [2]Helmholtz Center Munich Institute of Computational Biology, Munich, Germany. [3]Helmholtz Center Munich Institute of Functional Epigenetics, Munich, Germany. [4]TUM School of Life Sciences Weihenstephan, Technical University of Munich, Freising, Germany. [5]Department of Computer Science, TUM School of Computation, Information and Technology, Technical University of Munich, Garching, Germany. [6]German Centre for Cardiovascular Research (DZHK), Munich Heart Association, Partner Site Munich, Berlin, Germany. [7]These authors contributed equally: Corinna Losert, Matteo Zambon. ✉E-mail: antonio.scialdone@helmholtz-munich.de

of an objective and unbiased benchmarking framework for evaluating GRN algorithms using prior knowledge, an inherently difficult task given the algorithms' variety and the heterogeneity of their strategies.

Recent reviews on GRN inference have predominantly focused on the use of single-cell multiomics data (e.g., Badia-i Mompel et al, 2023), a pivotal approach in this field. However, this represents just one among many methodologies currently under development. A broader and more versatile area of GRN inference involves incorporating prior knowledge, which spans diverse strategies, including data-type-independent methods such as the use of topological priors and generalized graph priors (Nair, 2017; Stock et al, 2024). While many algorithms have been designed to leverage these approaches, a comprehensive review that systematically presents and compares these strategies is still lacking.

In this review, we provide an overview of prior-knowledge-informed GRN inference from scRNA-seq data from three perspectives. First, we categorize the available sources of prior knowledge and the strategies that algorithms use to incorporate them into the inference process. Secondly, we classify the algorithms based on their computational approaches, their ability to incorporate flexible graph priors and their capability to handle the specific features of scRNA-seq data. Based on this classification, we propose a framework for benchmarking algorithms using graph representations of prior knowledge, which offer flexibility in utilizing diverse sources of priors. This framework aims to disentangle the contributions of the prior knowledge and the algorithm itself to the overall performance, addressing the confounding factors present in current benchmarking studies. By emphasizing these three aspects of GRN inference with prior knowledge, this review is designed to address the needs of two primary audiences: researchers applying GRN inference algorithms and those interested in developing novel computational methodologies. For the former, we provide insights into the types of prior knowledge that current algorithms can handle, and which algorithms are most suitable based on their data. For the latter, we offer a comprehensive overview of the available computational strategies, emphasizing critical aspects and opportunities for combining approaches to enhance performance. By balancing practical application insights with detailed descriptions of algorithmic strategies, the review aims to serve as a valuable resource for both groups, reflecting its dual focus on application and innovation.

# Gene regulatory networks and their inference

## Gene regulatory networks

The deeper understanding of mechanistic processes in a cell is still a focus of ongoing research in systems biology. These processes are governed by a large network of interactions between the DNA, proteins, different RNAs and small molecules. They control cell proliferation and apoptosis, but can also help explain the stability of cell fate and other processes in each cell (Mukhopadhyay et al, 2020). The entirety of these regulatory interactions can be represented as a network (de Jong, 2002), which can be thought of as the fundamental organizational scheme for a cellular system.

Due to the high complexity of networks that describe all DNA, RNA, protein and small molecule interactions, the regulatory network is often decomposed into several different layers. The most fundamental one, based on gene expression, describes the binding of regulating proteins, called transcription factors (TFs), to the promoter of their regulated target genes (TGs) on the DNA. The resulting networks are also called transcription regulation networks, or TF-TG-Networks (Vázquez et al, 2004). This approach simplifies the perspective of understanding gene regulation by only focusing on transcriptional regulation, which describes the levels of transcription of a target gene depending on the regulation by a transcription factor, compared to a broader definition of a network that also includes other regulatory effects, such as epigenetic modifications (Erbe et al, 2023; Tejada-Lapuerta et al, 2023). Such a mechanistic model of regulatory interactions can, for instance, help predict how cells react to perturbations, such as drug treatments, which could be valuable for drug target identification and understanding therapeutic effects (Park et al, 2022). In addition, by identifying key regulators for specific cell types, it becomes possible to reprogram cells into different types, a prospect of high interest in regenerative medicine, where cell reprogramming could aid in tissue repair or the treatment of degenerative diseases (Aguirre et al, 2023). Other downstream analyses include identifying functional modules from the GRN, comparative analysis between multiple GRNs, and conducting in-silico perturbation experiments aimed at simulating the effects of potential interventions on cellular behavior (Badia-i Mompel et al, 2023). A TF-TG GRN is usually not strictly bipartite since transcription factors can also regulate other transcription factors. Therefore, we consider the resulting GRN as a homogeneous graph of interactions between different genes.

Recently, extended versions of GRNs that include regulatory elements (RE) as nodes have been constructed (Bravo González-Blas et al, 2023; Kamal et al, 2023). REs are regions on the DNA where TFs bind and regulate the expression of the target gene, such as promoters, enhancers, silencers, insulators and locus control regions. Since most approaches mainly focus on the incorporation of enhancers into the GRN, this kind of GRN is often called enhancer GRN (eGRN). It introduces multiple types of edges to the network, namely those going from transcription factors to the regulatory elements (TF-RE) and edges going from regulatory elements to the target genes (RE-TG). To compare them to other regular GRNs, a TF-TG network can be extracted from the eGRN in a post-processing step. In both TF-TG and eGRNs, the graph representation offers the advantage of leveraging theoretical frameworks and the wide range of analytical methods developed within the broader field of (biological) network science (Hetzel et al, 2021).

There are further characteristics that distinguish different types of GRNs. One example is resolution, where GRNs are either inferred from multiple cell types or individuals (Cha and Lee, 2020; Chasman and Roy, 2017), or at the highest resolution specifying personalized and cell type-specific GRNs (Bafna et al, 2023). Another feature is the size of the gene sets of interest, where some GRNs span as little as 5 genes (Atanackovic et al, 2023), whereas other GRNs include thousands of genes, which can be selected, for example, based on their highly variable expression levels (McCalla et al, 2023). In general, the larger the set of genes and the resolution of the GRN, the harder the inference problem becomes. To properly leverage the input scRNA-seq datasets, these should show

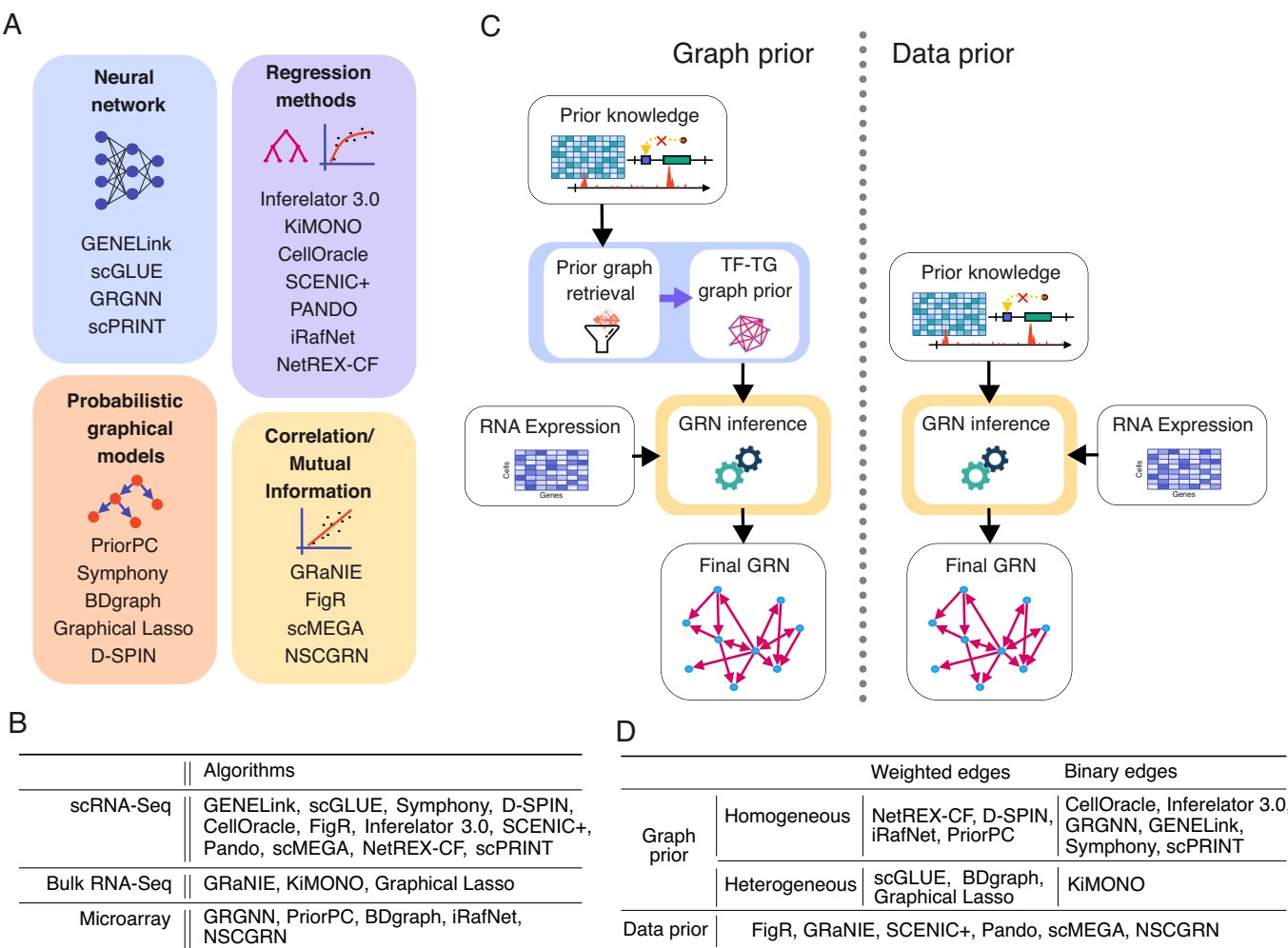

**Figure 1. Overview of available GRN inference algorithms leveraging different types of transcriptomic data and representations of prior knowledge.**

(A) Classification of GRN inference methods according to the type of algorithm they employ. (B) GRN inference algorithms that leverage prior knowledge were designed to work on different types of gene expression data (first column). Methods tailored for scRNA-seq data are expected to perform better on this data type because they address its unique challenges, such as sparsity and high intrinsic noise. (C) Schematic representation of the main steps of algorithms leveraging "graph-based" (left) or "data-based" (right) priors. With graph-based priors, the algorithms distill all prior knowledge in a TF-TG prior graph, which is then combined with gene expression data to generate a final GRN. The algorithms that use data-based priors, instead, combine prior knowledge and transcriptomic data directly during the inference to build a GRN, without generating any prior graph. (D) Classification of GRN inference methods based on the type of integrated prior.

high variability in the gene set of interest, which might, for example, be caused by different external conditions or perturbations (Aguirre et al, 2024; Saint-Antoine and Singh, 2023). However, some work suggests that the heterogeneity within scRNA-seq data from a single condition provides enough variability to infer meaningful regulatory relationships (Gupta et al, 2022). This review is not restricted to a specific resolution or size but addresses the general task of GRN inference.

## Types of inference algorithms

Various computational approaches have been applied to GRN inference, each leveraging different strategies and techniques. Broadly, GRN inference algorithms can be grouped into four categories based on their computational approach: correlation-based methods, regression methods, probabilistic graphical models, and neural network-based approaches (see Fig. 1A). Each of these

categories offers unique strengths and limitations, depending on the underlying assumptions and techniques employed.

Correlation- and mutual information-based methods rely on statistical dependencies to infer regulatory relationships between genes. FigR (Kartha et al, 2022) employs the non-parametric Spearman's rank correlation coefficient to detect monotonic relationships between gene expression levels and chromatin accessibility scores. In contrast, GRaNIE (Kamal et al, 2023) and scMEGA (Li et al, 2023b) use the linear Pearson's correlation coefficient for the same purpose. While correlation-based approaches are fast, scalable, and robust (Li et al, 2023b), they have notable limitations, particularly in detecting more complex non-linear relationships. To address this, NSCGRN (Liu et al, 2022) utilizes mutual information, which, however, cannot distinguish between positive and negative relationships. More generally, simple statistical measures struggle to capture complex causal interactions, often producing dense networks with numerous indirect edges that

can obscure true regulatory signals (Hawe et al, 2019). In addition, these methods cannot establish directed edges without prior knowledge of transcription factors.

Probabilistic graphical models (PGMs) use directed or undirected graphs to denote probabilistic relationships between variables. PriorPC (Ghanbari et al, 2015), for example, utilizes Bayesian networks, which encode conditional dependencies between variables as directed acyclic graphs, offering a more refined approach that reduces the prevalence of indirect edges. However, the assumption of acyclicity in Bayesian networks is a major limitation since it prevents the representation of feedback loops, a common feature in GRNs (Osella et al, 2011). Other PGM methods such as BDgraph (Mohammadi and Wit, 2019) or Graphical Lasso (Friedman et al, 2008), circumvent this issue by using undirected graphs. Both are based on Gaussian graphical models, which typically assume that the data follows a Gaussian distribution. This does not hold for single-cell RNA-seq data, which are over-dispersed and may be zero-inflated (Chen et al, 2018; Durif et al, 2019). To cope with this, BDgraph uses specific copula approaches to handle non-Gaussian data (Choudhary and Satija, 2022). Graphical Lasso (Friedman et al, 2008) focuses on sparsity, a well-known feature of GRNs, by introducing an L1-regularization penalty that enforces a sparser graph estimation. Importantly, both methods can incorporate priors on the graph structure either globally or for individual edges. Symphony (Burdziak et al, 2019) leverages a Bayesian hierarchical multi-view mixture model that allows the inference of cell type clusters and GRNs simultaneously. D-SPIN (Jiang et al, 2024) leverages spin networks, a type of maximum entropy model.

Regression-based approaches infer statistical relationships between genes using feature selection techniques. These techniques enable the detection of more complex dependencies and accommodate feedback loops and cycles, making them more versatile compared to simple correlation methods. The ability to model non-linear dependencies makes these methods particularly powerful for GRN reconstruction. Pando (Fleck et al, 2023) showcases a basic linear regression model. iRafNet (Petralia et al, 2015) employs random forest regression, a tree-based machine learning strategy. SCENIC+ (Bravo González-Blas et al, 2023) utilizes a gradient boosting algorithm, another ensemble method using decision trees. KiMONO (Ogris et al, 2021) applies LASSO-penalized regression to enforce sparser predictions. Inferelator 3.0 (Gibbs et al, 2022) tested three different regression approaches and achieved the best results with Adaptive Multiple Sparse Regression, a multitask learning approach that also enforces sparse predictions. CellOracle (Kamimoto et al, 2023) uses ridge regression, which adds an L2-regularization, either as Bayesian ridge regression or as Bagging ridge regression.

Neural network-based approaches, particularly deep learning models, leverage loss optimization via backpropagation to learn complex relationships from large data. GRGNN (Wang et al, 2020) employs Graph Neural Networks in a (semi-) supervised graph classification setting. GENELink (Chen and Liu, 2022) utilizes a Graph Autoencoder architecture in a self-supervised link prediction setting. scGLUE (Cao and Gao, 2022) combines the Graph Autoencoder with multiple variational autoencoders for the different data types, and adds a discriminator to align the different

latent spaces. scPRINT (Kalfon et al, 2024) uses a transformer architecture, training a foundation model on millions of single cells, and relies on the learned attention heads to predict the final GRN. One advantage of DL methods is that they do not rely on many assumptions about the underlying graph structure, which allows for improved performance compared to other methods (Shu et al, 2021; Yuan and Bar-Joseph, 2019). However, these approaches typically require significant computational resources, making them less practical for large-scale applications without extensive computational infrastructure.

Each of these categories of algorithms offers distinct advantages and challenges and requires specific strategies to incorporate prior knowledge. The choice of algorithm depends on the specific requirements of the GRN reconstruction task, such as data type, scale, and the desired balance between accuracy and computational efficiency.

All algorithms included in this review use gene expression data (Fig. 1B) combined with prior knowledge to infer GRNs. However, not all of them have been designed specifically for scRNA-seq data. Reconstructing GRNs from scRNA-seq data can potentially uncover new insights by modeling and exploiting the single-cell level heterogeneity. By modeling GRNs on a high-resolution cell type level or even per individual, it is possible to investigate context-specific differences in regulation between cell types or individuals. McCalla et al (McCalla et al, 2023) outline that GRN inference from one scRNA-seq dataset can reach the same performance as the GRN inference from multiple collected bulk RNA-seq datasets. At the same time, this data type poses new challenges, for example, due to its sparsity and noise (Akers and Murali, 2021). As a result, an algorithm designed for bulk sequencing data might need to be revised to work as efficiently on scRNA-seq data. A careful preprocessing of scRNA-seq data is also essential for its use in GRN inference. The raw output of RNA-seq experiments is read sequences that have to be aligned to the respective genome, resulting in a count score per annotated gene. Standard preprocessing steps include quality control and count normalization, followed by feature selection, batch integration and potentially further dimensionality reduction (Heumos et al, 2023). The quality control includes the correction for ambient RNA, the filtering of low-quality libraries, and the detection and removal of doublets. Recommended normalization techniques depend on the downstream task; an overview of different methods is available in a recent comparison of Ahlmann-Eltze and Huber (2023). The feature selection is usually done by subsetting the genes of interest to the highly variable genes. For batch integration, several algorithms including linear-embedding models like Harmony (Korsunsky et al, 2019) and Scanorama (Hie et al, 2019), and deep-learning approaches such as scANVI (Xu et al, 2021), scVI (Lopez et al, 2018), and scGen (Lotfollahi et al, 2019) were recently benchmarked by (Luecken et al, 2022). Finally, dimensionality reduction techniques like Principal Components Analysis, t-SNE (van der Maaten and Hinton, 2008), UMAP (McInnes et al, 2020) and PHATE (Moon et al, 2019) can be applied for visualization purposes. Algorithms presented in this study assume already preprocessed scRNA-seq data as an input since the exact choice of preprocessing depends on the specific dataset and thus cannot be easily automated.

# Leveraging prior knowledge for GRN inference

As discussed above, GRN inference typically relies on measured gene expression data obtained from bulk RNA-seq or scRNA-seq experiments. When additional data beyond gene expression is incorporated to support the inference process, the algorithms are regarded as leveraging prior knowledge. Below, we detail the various types of prior knowledge, the sources from which they can be derived, and the algorithms capable of integrating these different types of priors.

## Transcription factor databases

One commonly available source of additional information is the list of transcription factors in the model organism. This information can be used to filter out interactions that do not involve at least one TF and to infer the directionality of regulation based on co-expression patterns. Lists of TFs can be obtained from various resources, depending on the organism under study. For example, TF lists can be derived from gene annotation databases such as Ensembl (Harrison et al, 2024) or Uniprot (The UniProt Consortium, 2023), where genes annotated as TFs or associated with the relevant Gene Ontology terms can be identified. Dedicated TF databases, like PlantTFDB (Jin et al, 2017) for plants or AnimalTFDB (Shen et al, 2022) for animals, provide organism-specific collections of TFs. Alternatively, experimental evidence-based databases, such as JASPAR (Rauluseviciute et al, 2024) or the Encyclopedia of DNA elements (The ENCODE project consortium, 2012), often include ChIP-seq data and other experimental evidence of TF binding. These resources not only supply TF lists but also additional regulatory information that can be leveraged by GRN inference algorithms, as described below in further detail. A simple TF list, classifying genes as TFs, is widely used by many state-of-the-art algorithms, such as GENIE3 (Huynh-Thu et al, 2010). This review focuses exclusively on algorithms that incorporate at least one additional source of prior knowledge, leveraging these resources to enhance the inference process.

## Experimental sources of prior knowledge

Most of the prior knowledge that can be integrated into the GRN inference process is derived from biological experiments. Here, we outline the different types of experimental prior knowledge, the experimental protocols with which those priors are built, the computational strategies that are applied to incorporate them and their specific characteristics and limitations. In Table 1, we summarize the most common sources of experimental prior knowledge.

Very often, the algorithms leverage multiple prior sources together. The most common combination is to use chromatin accessibility data from (single-cell) ATAC-seq and TF-binding motif enrichment. In this case, the regulatory elements inferred from accessibility peaks are then linked with matching TF motif enrichment. Methods that leverage both accessibility and motif enrichment data are Inferelator 3.0, FigR, SCENIC+, GRaNIE, scMEGA, Pando, Symphony, and CellOracle. Other algorithms, instead, leverage data from knockout experiments and protein–protein interaction data (iRafNet) from binding motifs,

knockout experiments and ChIP-seq data (NetREX-CF), or from an arbitrary number of unpaired data modalities, like scGLUE or KiMONO. Hawe et al (2022) construct a prior network from a combination of protein–protein interactions, transcription factor and histone ChIP-seq, eQTL and SNP data to then run BDgraph, Graphical Lasso and iRafNet with the prior network.

As often there is no ground truth available for validating the resulting GRNs, experimental data that can be integrated as prior knowledge is also, at the same time, the best resource for validating inferred links. For example, data from knockout experiments is used for prior construction in some methods (iRafNet, NetREX-CF), whereas it is used for validation of inferred GRN links in others (SCENIC+, GRaNIE).

In this section, we discuss each experimental source by first outlining the primary goal of the experiment, such as identifying regulatory elements or specific types of interactions. This is followed by a brief explanation of the experiment and its resulting data output. We then describe how this data is currently utilized in published GRN inference algorithms. Lastly, we highlight any limitations or provide additional insights where relevant.

### Chromatin accessibility data (ATAC-seq)

Chromatin accessibility can provide information on transcription factor binding sites and the regulatory potential of a genetic locus. Therefore, it can help identify regulatory elements (mainly enhancers) as GRN nodes that, in turn, link the transcription factors to their target genes.

Bulk or single-cell ATAC-seq (Buenrostro et al, 2015) is one of the most commonly used experimental techniques to quantify chromatin accessibility, even though other assays like DNase-seq or FAIRE-seq could also be used. In an ATAC-seq (Assay for Transposase-Accessible Chromatin using sequencing) experiment, nuclei are treated with a transposase enzyme (e.g., hyperactive Tn5 transposase), which inserts sequencing adapters into accessible DNA regions. This is followed by sequencing. The locations with a high amount of sequencing reads (peaks) represent the regions of increased accessibility.

The candidate REs identified from peaks in ATAC-seq data are usually linked to potential target genes by calculating whether the peak overlaps with a pre-specified region around the target gene (see the section "Genomic distance"). If paired data is available, where RNA-seq and ATAC-seq are performed on the same single cells (Chen, 2019; Ma et al, 2020), the peaks might be associated with target genes by computing the correlation between chromatin accessibility at the peak and target gene expression or by more sophisticated methods like tree-based regression or others (e.g., GRaNIE, Pando, SCENIC+, FigR, scMEGA). For the RE-TG links based on the correlation of chromatin accessibility and RNA expression, GRaNIE filters for positive correlation, since negative correlation does not have a biological explanation and thus is considered noise (Kamal et al, 2023). In the same way, TFs also might be linked to the regulatory regions by associating the transcription factor expression to the peak accessibility (e.g., FigR, GRaNIE). Another way to link TFs to REs is to combine the chromatin accessibility data with binding motif data, which will be outlined in more detail in the next section "TF-Motif Enrichment".

As especially single-cell ATAC data is usually very sparse, most GRN inference methods pre-process the data to reduce sparsity (e.g., by aggregating the data at the cell-type level) before

**Table 1.**   Overview of experimental data ("Source") from which prior knowledge ("Prior") useful for GRN inference can be extracted.

| Source | Prior knowledge extracted | Input | Validation |
|---|---|---|---|
| Chromatin accessibility (sc/bulk ATAC-seq) | RE RE-TG TF-RE | Inferelator 3.0, scGLUE, CellOracle, Symphony GRaNIE, Pando, SCENIC+, FigR, scMEGA FigR, GRaNIE | |
| TF-Motif enrichment (DNA sequencing) | TF-RE | Inferelator 3.0, CellOracle, scMEGA, Pando, scMEGA, SCENIC+, GRaNIE, FigR,Symphony, NetREX-CF | |
| Protein-DNA binding (TF ChIP-sequencing) | TF-RE | scGLUE, NetREX-CF | *SCENIC+,scPRINT,GRaNIE,FigR,GRGNN,iRafNet,GENELink,D-SPIN,CellOracle* |
| Perturbation/Knockout Experiments (sc/bulk RNA expression) | TF-TG | iRafNet, NetREX-CF, D-SPIN | *SCENIC+,GRaNIE,scPRINT,GENELink* |
| Protein–protein interaction networks | TF-TG | iRafNet | |
| Protein-DNA binding (Histone modification ChIP-sequencing) | RE RE-TG TF-RE | GRaNIE | *SCENIC+,Pando* |
| Enhancer activity essays (STARR-seq) | RE | | *SCENIC+* |
| DNA sequence conservation | RE | Pando | *GRGNN,iRafNet* |
| Genomic distance | RE-TG | scMEGA, NetREX-CF, scGLUE, Symphony,KiMONO, SCENIC+, CellOracle, Pando, GRaNIE, Inferelator 3.0, FigR | |
| Chromatin interaction (Hi-C) | RE-TG | GRaNIE, scGLUE | *SCENIC+* |
| Expression quantitative trait loci (eQTLs) | RE-TG | scGLUE | *KiMONO,GRaNIE* |
| Intrinsic feature extraction (sc/bulk RNA expression) | TF-TG | GRGNN, iRafNet, NetREX-CF | |

The prior knowledge can consist of a list of regulatory elements (RE), links between regulatory elements and target genes (RE-TG), links between transcription factors and regulatory elements (TF-RE), and links between transcription factors and target genes (TF-TG). The table also includes a list of algorithms that leverage each type of prior extracted from a given type of data. These algorithms can then utilize this information either as input ("Input" column) to enhance GRN inference or as a way to evaluate their results ("Validation").

computing the correlation between chromatin accessibility and gene expression. With the recent emergence of paired scATAC- and scRNA-seq datasets, the performance of these methods could be further improved compared to computationally paired GRN reconstruction. SCENIC+, scMEGA and scGLUE already showcase the application on a 10x Genomics multiome dataset, where RNA expression and chromatin accessibility are simultaneously assessed.

While leveraging chromatin accessibility data has been shown to improve GRN inference (Alanis-Lobato et al, 2023; Argelaguet et al, 2022), one major limitation to consider is that chromatin accessibility does not always imply activity. Thus, it can be misleading when used to derive information about TFs, REs, and TGs interactions (Kamal et al, 2023).

### TF-Motif Enrichment

The use of transcription factor binding motifs is another potential prior that can be used to build links between TFs and TGs via REs.

The binding motifs of TFs are usually collected in motif-binding databases. Those databases differ by whether they include binding motifs based only on experimental data from ChIP-seq, DNase-seq,

or SELEX (e.g., JASPAR (Rauluseviciute et al, 2024), TRANFAC (Matys et al, 2006)) or whether they also include computational predictions of motif-binding sites (e.g., HOCOMOCO (Kulakovskiy et al, 2018)). Usually, they contain position frequency matrices that summarize the occurrence of each nucleotide at each position in a set of experimentally observed TF-DNA interactions. Those position frequency matrices can then be transformed into position weight matrices (PWMs) via probabilistic or energistic models.

To link TFs to TGs via motif databases, in the first step, a set of candidate REs needs to be identified. This can be done, for example, by choosing a fixed region upstream and downstream of the transcription start site of the target genes (see the section "Genomic distance"), as in Inferelator 3.0 or NetREX-CF. Alternatively, REs can be inferred first from other types of data (e.g., ATAC-seq, DNase-seq, ChIP-seq, Hi-C), or from databases, like ENCODE (Fleck et al, 2023; Gibbs et al, 2022; The ENCODE project consortium, 2012). Once the candidate REs are identified, a motif enrichment analysis is performed to infer potential links between them and TFs. In many GRN inference algorithms, this is done by using the R package motifmatchr (Schep, 2024) in combination

with chromVar (Schep et al, 2017) to calculate the enrichment scores of motifs in accessible regions (e.g., Pando, scMEGA, FigR). The Inferelator 3.0 method (Gibbs et al, 2022) was tested with different motif databases (CisBP, JASPAR, TRANSFAC) and it was shown that the employed motif library significantly affects the network output, but none of the databases was shown to outperform the others. In the SCENIC+ (Bravo González-Blas et al, 2023) method, this ambiguity is addressed by building a consensus position weight matrix from more than 40,000 PWMs from 29 collections of binding motifs to generate the most extensive TF binding motif database to date containing about 1.5k human, 1.3k mouse, and 467 drosophila TFs. The authors showed that keeping the diversity of different motifs for a TF results in a higher accuracy than having only one archetype motif per TF. The evaluation of three different motif enrichment methods further showed that cisTarget and DEM outperform Homer.

Leveraging TF binding motifs knowledge for refining TF-TG links can help distinguish direct from indirect targets and define the directionality of gene-gene interactions for GRN inference. However, TF motifs can also be very similar across TFs, and in general, motif-derived prior knowledge GRNs are noisy and influenced by choice of the motif library (Gibbs et al, 2022).

### Protein-DNA binding assays (TF ChIP-seq)

Similarly to ATAC-seq data, ChIP-seq data can reveal potential interactions between TFs and REs by identifying potential TF binding sites.

With ChIP-seq (Chromatin Immunoprecipitation followed by sequencing) experiments, it is possible to identify DNA regions bound by specific proteins, such as transcription factors or histones. In these experiments, chromatin is cross-linked in the cell to stabilize the interactions between the TF and the DNA. Then the DNA is fragmented and a specific antibody that binds to the TF is added. Subsequently, the antibody-TF-DNA complexes are immunoprecipitated, the DNA is purified and the bound DNA fragments are sequenced. By analyzing the sequencing data, regions in the DNA where the TF was binding can be identified (ChIP-seq peaks). From this, TF-RE links can be inferred. Databases like ENCODE (The ENCODE project consortium, 2012), Remap (Hammal et al, 2022) or Unibind (Puig et al, 2021) collect and process information from ChIP-seq experiments to provide transcription factor binding sites predictions.

ChIP-seq data can be used as prior in different ways. NetREX-CF (Wang et al, 2022) generates TF-TG priors by combining TF-RE connections from TF ChIP-seq with RE-TG connections from a genomic distance prior (see the section "Genomic distance"), by assuming that a RE-TG link occurs whenever the RE falls within a gene body or up to 1 kb upstream of the gene. scGlUE (Cao and Gao, 2022) leverages ChIP-seq data from ENCODE in combination with ATAC-seq data to establish a ranking of the most likely target genes for each TF by overlapping ChIP-seq peaks with ATAC-peaks. Methods like GRaNIE (Kamal et al, 2023) and PriorPC (Ghanbari et al, 2015) outline the possibility of using ChIP-seq data priors as an input to the algorithm to identify regulatory regions, but do not showcase it in their applications.

TF ChIP-seq data was also used to build curated ground truth GRNs, as in the DREAM5 challenge yeast dataset (Marbach et al, 2012), which is used by GRGNN (Wang et al, 2020) and iRafNet (Petralia et al, 2015), and the curated ground truths included in the

benchmarking analysis BEELINE (Pratapa et al, 2020), which are used, for example, by GENELink (Chen and Liu, 2022). Such curated ground truth networks are split by GRGNN and GENELink into a training part that serves as a prior and a validation part that is used to validate the predictions. Similarly, scPRINT (Kalfon et al, 2024) uses an intersection of perturbation- and ChIP-seq-based ground truth network from McCalla et al. (McCalla et al, 2023) for validation. Also in FigR, GRaNIE, CellOracle, D-SPIN and SCENIC +, ChIP-seq data is used for validating the results of the GRN inference.

The use of ChIP-seq to identify TF binding sites is limited by the lack of high-quality antibodies for some TFs, and the need for large amounts of homogeneous cells (e.g., cell lines, bulk tissues). Experimentally mapping TF binding sites on tissues with a high diversity like, for example, the brain, remains challenging (Bravo González-Blas et al, 2023).

### Knockout/perturbation experiments

Knockout and, more in general, perturbation experiments enable the discovery of TF-TG interactions by screening for genes that are affected following the perturbation of specific transcription factors (Petralia et al, 2015).

One example of such experiments is CRISPR-based Perturb-seq assays (Replogle et al, 2022), as used by D-SPIN (Jiang et al, 2024). In these experiments, a library of guide RNAs is designed to target specific genes (in this case TFs) to knockout (CRISPR/Cas9), knockdown (CRISPRi) or activate (CRISPRa) the transcription of the gene. These guide RNAs are then inserted into the single cells, which subsequently leads to a cellular response to the perturbation. Subsequently, scRNA-seq is utilized to measure the impact of the perturbation on gene expression. By performing differential expression analysis between perturbed and unperturbed cells, the TGs influenced by the perturbation of the TF can be identified. Notably, perturbations are not restricted to genetic mutations; they can also involve drug-based interventions that disrupt diverse cellular mechanisms (Peidli et al, 2024). Simulations have demonstrated that perturbation data enhance the ability to distinguish direct gene-gene interactions more effectively. In addition, these perturbations introduce extra variability into expression data, which can be harnessed to improve the accuracy of GRN inference (Aguirre et al, 2024; Saint-Antoine and Singh, 2023). A large amount of knockdown experiment data is, for example, provided by ENCODE (The ENCODE project consortium, 2012).

Several algorithms utilize perturbation data to construct priors that are incorporated into GRN inference. iRafNet (Petralia et al, 2015) integrates the information of knockdown experiments provided within the DREAM5 (Marbach et al, 2012) challenge as prior knowledge by calculating a weight for each transcription factor to target interaction based on the p-value resulting from a t-test run on the expression levels of a given gene before and after the TF knockout. Similarly, NetREX-CF (Wang et al, 2022) also infers prior information on network edges by considering as potential target genes only those with a statistically significant difference in expression level between the control and the perturbed condition.

Other methods use the results of perturbation experiments to validate the inferred interactions (GRaNIE, SCENIC+). SCENIC+ calculates the gene set enrichment of the inferred targets for each

TF in relation to the differentially expressed genes observed in perturbation experiments of the same TF, using perturbation data from the ENCODE (The ENCODE project consortium, 2012) database. As mentioned already in the TF ChIP-seq section, scPRINT (Kalfon et al, 2024) validates the predicted GRN against the intersection of ChIP-seq- and perturbation-based ground truth provided by (McCalla et al, 2023). By leveraging the curated ground truths by BEELINE (Pratapa et al, 2020), GENELink (Chen and Liu, 2022) also makes use of loss-of-function experiments.

While perturbation experiments provide valuable insights into gene regulation, the availability of such data remains limited due to the high cost, time investment, and cell-type specificity of these experiments.

### Protein–protein interaction networks

Protein–protein interaction (PPI) networks represent another type of experimentally derived prior, with many potential TF-TG interactions curated in large databases. These databases typically report physical interactions between protein pairs and sometimes functional interactions as well. A notable example is the Bioplex interactome (Huttlin et al, 2021), which was generated using Affinity-Purification Mass Spectrometry on human cell proteins. Another widely used example is HuRI (Luck et al, 2020), a reference interactome map comprising approximately 53,000 human binary protein interactions, created through systematic yeast two-hybrid assays.

iRafNet (Petralia et al, 2015) incorporates TF-TG interactions included in PPI networks as priors, applying a diffusion kernel transformation to pre-process the PPI data. While PriorPC (Ghanbari et al, 2015) also acknowledges the potential of PPIs as priors, it does not showcase any application. (Hawe et al, 2022) uses PPIs from BioGrid (Oughtred et al, 2019) to construct gene-gene edges that are then used as an input to the BDgraph (Mohammadi and Wit, 2019) and Graphical Lasso (Friedman et al, 2008) algorithms.

### Protein-DNA binding assays (histone modification ChIP-seq)

Histone modifications such as H3K27ac are well-established markers of active regulatory elements (Creyghton et al, 2010). Consequently, assessing the presence of this modification via ChIP-seq serves as a valuable prior for selecting potential REs. In this case, in the ChIP-seq experiment, the fragmented chromatin is incubated with an antibody specific to H3K27ac (see ChIP-seq section "Protein-DNA binding assays (TF ChIP-seq)") to identify the regions with high levels of this histone modification.

For example, the authors of GRaNIE (Kamal et al, 2023) suggest using this kind of data as an alternative to ATAC-seq. By correlating the H3K27ac mark with RNA expression data, GRaNIE infers TF-RE and RE-TG interactions. Similarly, SCENIC+ (Bravo González-Blas et al, 2023) employs H3K27ac ChIP-seq for validating identified active regulatory regions. The authors of Pando (Fleck et al, 2023) also used a related method (Cut&Tag; (Kaya-Okur et al, 2019)) to validate their inferred regulatory regions by assessing the levels of H3K27ac modifications.

### Enhancer activity assays

Another method to identify REs like active enhancers is massively parallel reporter assays (MPRAs). MPRAs are a powerful tool used to evaluate how DNA sequences, particularly regulatory elements like enhancers and promoters, influence gene expression. In an MPRA experiment, a library of DNA sequences is designed to include candidate REs, mutated variants, and randomized controls, each tagged with a unique barcode for identification. These sequences are cloned into reporter constructs that contain a minimal promoter and a reporter gene, and then introduced into cells. Inside the cells, the regulatory sequences drive the expression of the reporter gene. RNA sequencing is subsequently performed to quantify the barcode abundance, providing a direct link between the activity of each sequence and its influence on gene expression. The STARR-seq protocol, a specific type of MPRA (Arnold et al, 2013) designed for enhancer identification, utilizes high-throughput sequencing to quantify the activity of candidate enhancers. SCENIC+ (Bravo González-Blas et al, 2023) uses STARR-seq data to validate the regulatory regions identified by the algorithm.

### Conservation of DNA sequences

After an initial list of candidate REs is identified, for instance through chromatin accessibility data, further filtering can be performed based on the evolutionary conservation of DNA sequences. This approach, as used by Pando (Fleck et al, 2023), involves overlapping the candidate REs with a set of highly conserved DNA regions across 30 mammalian species. The resulting list of candidate REs is considered more reliable due to their conservation over evolution.

Similarly, the curated yeast network used for the DREAM5 challenge (Marbach et al, 2012) applies evolutionary conservation as a key criterion to refine the list of TF binding sites derived from ChIP-seq data. This curated network serves as a benchmark for validating the iRafNet algorithm (Petralia et al, 2015) and is also utilized by GRGNN (Wang et al, 2020), both as a prior and for validation purposes.

### Genomic distance

Genomic distance is a widely used and readily available metric for constructing priors on the relationships between candidate REs and TGs. REs, often identified from ATAC-seq data, are typically associated with regions around the transcription start site (TSS) of target genes. Depending on the method, the overlap window for linking REs to TGs ranges from 1 kb upstream of the TSS (Wang et al, 2022) to 250 kb (Kamal et al, 2023), covering potential cis-regulatory elements. Inferelator defines the regulatory region as 200 bp upstream and 50 bp downstream of the TSS, while Symphony (Burdziak et al, 2019) simply assigns each ATAC-seq peak to the nearest gene. Several methods that link regulatory regions to target genes include scGLUE, GRaNIE, SCENIC+, FigR, Pando, scMEGA, and CellOracle. NetREX-CF (Wang et al, 2022) associates candidate transcription factor binding sites and TF ChIP peaks with target genes by considering the gene body plus 1 kb upstream of the TSS. KiMONO (Ogris et al, 2021) uses genomic distance to link SNPs to methylation peaks and methylation peaks to genes. In addition, scGLUE (Cao and Gao, 2022) integrates genomic distance with Hi-C data, as discussed in the following section.

### Chromatin interactions (Hi-C)

Another source of experimental data that can be used to derive potential links between REs and TGs is data from Hi-C experiments.

This chromosome conformation capture technology measures the frequency at which two DNA fragments physically interact in the 3D space and can be valuable for detecting long-range gene regulation.

The scGLUE algorithm (Cao and Gao, 2022) integrates data from promoter capture Hi-C (pcHi-C), along with other priors, measured in 17 human primary blood cell types to calculate a weight defining pcHi-C-supported interactions between REs (peaks) and TGs. In this application, pcHi-C support for a candidate RE-TG pair was determined based on three criteria: (1) the gene promoter must be located within 1 kb of a bait fragment; (2) the peaks must be located within 1 kb of another-end fragment; and (3) significant contact between the bait and other-end fragment must be observed. In addition to scGLUE (Cao and Gao, 2022), the integration of chromatin conformation data for prior construction is also supported by GRaNIE (Kamal et al, 2023), which uses this data to define RE-TG pairs for further testing, although no specific application is provided in the paper. SCENIC+ (Bravo González-Blas et al, 2023) employs this data to validate inferred RE-TG links.

### Expression quantitative trait loci (eQTLs)

eQTL data is a valuable resource for identifying potential links between REs and TGs. eQTL data is typically obtained from experiments that measure both gene expression (e.g., via RNA-seq) and genomic variation. These studies assess the impact of single nucleotide polymorphisms (SNPs) on gene expression, often employing statistical models such as linear regression, where gene expression serves as the dependent variable and SNP genotype as the independent variable. When an SNP shows a statistically significant association with expression variation, it is classified as an eQTL, establishing a link between specific REs and their TGs. This approach helps identify genetic variants that influence gene expression through regulatory mechanisms. One resource that collects eQTL data across multiple human tissues is the Genotype-Tissue Expression (GTEx) project (The GTEx Consortium, 2020). With the increasing availability of single-cell data, recent efforts have focused on mapping cell-type-specific eQTLs, which could serve as valuable prior knowledge (Perez et al, 2022; van der Wijst et al, 2018; Yazar et al, 2022). The sc-eQTLGen Consortium (van der Wijst et al, 2020) was established to perform large-scale, systematic eQTL mapping across a wide range of single-cell datasets. In addition to mapping potential RE-TG links, a specific type of eQTL mapping, known as co-expression QTL mapping, can reveal potential TF-TG relationships by linking upstream TF regulators to downstream TGs (Li et al, 2023a).

The links between REs and TGs identified by eQTL mapping can be integrated as prior for RE-TG interactions. For instance, scGLUE (Cao and Gao, 2022) leverages eQTL data to identify peak-target gene interactions by verifying whether a peak derived from ATAC-seq data overlaps with an eQTL locus, and that locus is associated with the expression of the target gene. The eQTL data utilized by scGLUE is from the GTEx project (The GTEx Consortium, 2020). Other algorithms such as KiMONO (Ogris et al, 2021) and GRaNIE (Kamal et al, 2023) use eQTL data to validate predictions.

### Intrinsic feature extraction (RNA-seq)

Besides using additional data in combination with gene expression data, which we consider leveraging prior knowledge in this review,

there is also the possibility to enrich the inference by using algorithms to extract additional information from the same expression data. This process can also be seen as a kind of feature crafting from the expression data. An example of this approach is applying existing non-prior-based GRN inference methods, such as mutual information, Pearson correlation, Spearman Rank, or GENIE3 (Huynh-Thu et al, 2010), to gene expression data. The resulting graphs are then utilized as prior knowledge for GRN inference on either the same or a different set of gene expression data. This strategy is exemplified by the GRGNN algorithm (Wang et al, 2020), which constructs a 'noisy starting skeleton' using mutual information or correlation metrics to represent the relationships between gene expression profiles. NetREX-CF (Wang et al, 2022) constructs a co-expression network in order to extend the prior network to span more of the transcription factors. Another example is the use of multiple RNA-seq datasets as a time series, or the inference of pseudo-time (Matsumoto et al, 2017; Qiu et al, 2020; Singh et al, 2024) or RNA velocity (Singh et al, 2024) from the expression data, which can then be used as a prior. For example, iRafNet (Petralia et al, 2015) integrates time-series data and links two genes, $i$ and $k$, in the prior graph if the past values of gene $k$ are predictive of future values of gene $i$.

### Further experimental priors

In addition to the most commonly used prior sources as outlined in Table 1, KiMONO and scGLUE are designed to handle further input data in the form of a feature matrix. KiMONO (Ogris et al, 2021) mentions protein expression data, which can be added to the prior graph by establishing links to genes and other proteins via protein annotation data. KiMONO further mentions methylation, SNP and mutation data to extend the heterogeneous prior graph by leveraging the genomic distance as a prior to link these additional features to genes. scGLUE (Cao and Gao, 2022) similarly leverages snmC-seq DNA-methylation data. Hawe et al (2022) extend the genomic entities to CpG sites and construct CpG-to-gene priors using histone mark combinations (measured by ChIP-seq) and genomic distance.

## Topological prior

GRNs are characterized by specific structural properties, which can be quantified by topological measurements (Zhivkoplias et al, 2022). Recent studies showed that current GRN inference algorithms produce networks that lack the typical topological properties of GRNs, which can be important in predicting various biologically relevant properties (Stock et al, 2024), such as robustness to perturbations, etc. A common characteristic of GRNs is their scale-free node-degree distribution (Zhivkoplias et al, 2022). This implies that the node degree follows a power-law distribution, where a small number of nodes exhibit exceptionally high connectivity, serving as hubs with numerous interactions. Such hub genes can represent master regulators of a biological process.

To enhance accuracy, some algorithms leverage known GRN properties to establish topological priors and introduce constraints on graph topology. For example, constraints on the node-degree distribution are employed in certain methods (Nair, 2017). The BDgraph algorithm (Mohammadi and Wit, 2019) takes a related approach by using priors such as the expected sparsity coefficient, the G-Wishart distribution for the null adjacency matrix, and the

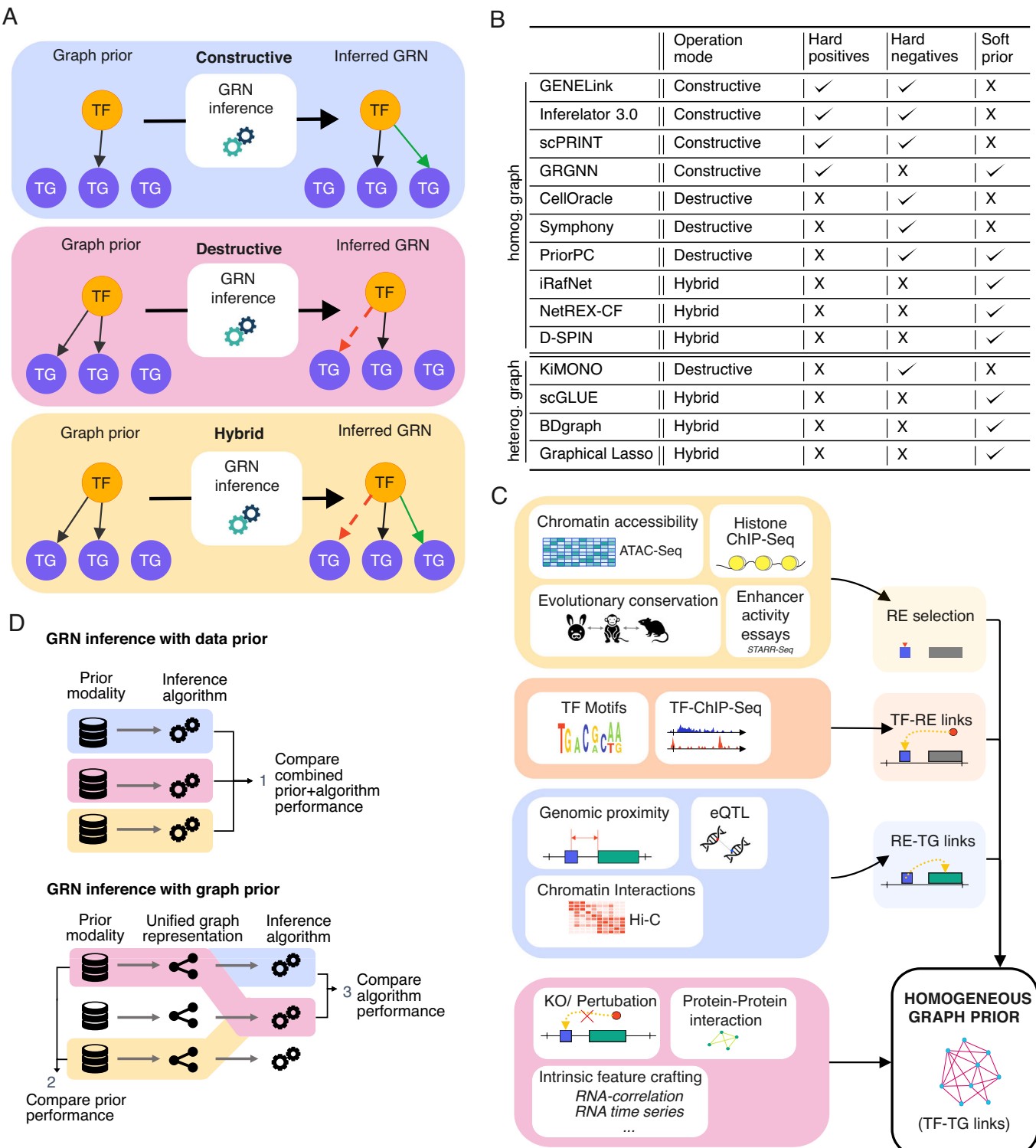

| | | Operation mode | Hard positives | Hard negatives | Soft prior |
|---|---|---|---|---|---|
| homog. graph | GENELink | Constructive | ✓ | ✓ | X |
| | Inferelator 3.0 | Constructive | ✓ | ✓ | X |
| | scPRINT | Constructive | ✓ | ✓ | X |
| | GRGNN | Constructive | ✓ | X | ✓ |
| | CellOracle | Destructive | X | ✓ | X |
| | Symphony | Destructive | X | ✓ | X |
| | PriorPC | Destructive | X | ✓ | ✓ |
| | iRafNet | Hybrid | X | X | ✓ |
| | NetREX-CF | Hybrid | X | X | ✓ |
| | D-SPIN | Hybrid | X | X | ✓ |
| heterog. graph | KiMONO | Destructive | X | ✓ | X |
| | scGLUE | Hybrid | X | X | ✓ |
| | BDgraph | Hybrid | X | X | ✓ |
| | Graphical Lasso | Hybrid | X | X | ✓ |

Bernoulli distribution for link inclusion. Another notable method is Graphical Lasso (Friedman et al, 2008), which promotes sparsity by applying a lasso penalty to the inverse covariance matrix. In addition, NSCGRN (Liu et al, 2022) incorporates commonly recurring GRN motifs as structural priors, enforcing local patterns like cascades and feedforward loops in the inferred networks.

In their preprint (Aguirre et al, 2024) recently summarized additional common attributes of GRNs, such as modular structure (gene modules), bidirectional and overall sparsity, as well as asymmetric degree distributions (in-compared to out-degree) that could be leveraged as further structural priors for directed GRNs.

**Figure 2.  Algorithms leveraging homogeneous graph priors enable disentangling the contributions of priors and inference methods in benchmarking studies.**

(A) Classification of GRN inference methods that leverage a homogeneous graph prior knowledge in terms of their operation mode. Constructive algorithms (top panel) add edges to the prior network, destructive algorithms (middle panel) remove edges from the prior network, and hybrid algorithms (bottom panel) can add and remove edges simultaneously. (B) This panel displays the operation mode of each algorithm, with an indication of whether they can incorporate hard positives, hard negatives, and soft prior. (C) Schematic illustration of the different approaches found among the reviewed methods to construct a homogeneous graph prior before the inference process that incorporates the RNA expression data. Some approaches directly infer TF-TG link priors (bottom panel). Other approaches build an enhancer GRN including regulatory elements first and then extracting the TF-TG graph from it. In this case, some priors are leveraged for selecting candidate REs ("RE selection"), for the definition of TF-RE links and for the definition of RE-TG links. (D) Benchmarking of GRN inference with prior knowledge. The upper panel shows a standard benchmarking setup, where the combination of a specific prior modality and algorithm are evaluated against each other. In this scenario, the effect of the prior and algorithm cannot be distinguished. The lower panel shows the two additional options when benchmarking methods with a unified graph prior input. This enables a comparative evaluation of the contribution of priors and algorithms separately.

## Direct data prior incorporation and graph prior extraction

In GRN inference, the way prior knowledge is incorporated can vary depending on the algorithm but generally falls into two main approaches. Some algorithms use prior knowledge to construct a graph, which serves as a "prior graph" during the inference process. In contrast, other methods integrate prior knowledge directly during the inference without building any prior graph first. We categorize the first approach as "graph-based" priors and the second as "data-based" priors (see Fig. 1C).

Algorithms employing "data-based" priors frequently combine transcriptomic and epigenomic data, especially in single-cell contexts (e.g., FigR, GRaNIE, SCENIC+, Pando, scMEGA). These typically establish correlations between chromatin accessibility at regulatory regions, such as promoters and enhancers, and gene expression. For instance, FigR, GRaNIE, SCENIC+, Pando, and scMEGA all construct enhancer-based GRNs, where TF-RE and RE-TG links are inferred first. Then, a TF-to-TG network is derived through additional post-processing steps. Other algorithms in this category incorporate different types of prior knowledge, such as NSCGRN (Liu et al, 2022), which uses topological priors on GRNs (see section above).

On the other hand, algorithms that utilize "graph-based" priors follow a two-step process. First, a prior graph is constructed, either using additional data sources (e.g., Inferelator 3.0, iRafNet, CellOracle, scGLUE, NetREX-CF, KiMONO, Symphony, BDgraph, Graphical Lasso, D-SPIN) or by downsampling curated ground truth GRNs (e.g., GENELink, PriorPC, GRGNN, scPRINT). In the second step, this prior graph serves as input for inferring the final GRN.

A key advantage of this approach is its flexibility. The second step can accommodate various types of priors because the prior graph is presented in an abstract, unified format. This allows researchers to construct the prior graph from diverse experimental data sources, unlike algorithms that rely on data-based priors, which require specific experimental inputs. In addition, the two-step process enables benchmarking studies to independently evaluate the contributions of performance gains due to the prior from the gains due to the actual inference algorithm.

The following sections provide an overview of the various subtypes of "graph-based" priors and the algorithms designed to operate with this type of input.

## GRN inference with graph priors

Graph-based priors are commonly represented using adjacency matrices $B$, where each entry $b_{i,j}$ denotes the weight of the edge

between nodes $i$ and $j$. In certain algorithms (e.g., CellOracle, GRGNN, GENELink, Inferelator 3.0, Symphony, scPRINT, KiMONO), these weights are binary, representing merely the presence or absence of a link. In other cases, the weights encode a confidence score, quantifying the uncertainty of the link, often due to inconsistencies across different sources of prior knowledge (e.g., PriorPC, iRafNet, NetREX-CF, scGLUE, BDgraph, Graphical Lasso, D-SPIN).

Graph priors can be categorized as either "homogeneous" or "heterogeneous", depending on their composition. Homogeneous graph priors consist solely of TF-TG interactions, as utilized in algorithms such as Inferelator 3.0, PriorPC, iRafNet, GENELink, CellOracle, GRGNN, Symphony, NetREX-CF, D-SPIN and scPRINT. In contrast, heterogeneous graph priors incorporate interactions with additional elements, such as REs in scGLUE or proteins in KIMONO. Hawe et al (2022) applies BDgraph, Graphical Lasso, and iRafNet to a heterogeneous graph prior. However, since iRafNet was originally presented to operate with homogeneous networks, we classify it under the homogeneous prior category. Figure 1D provides a detailed overview of the types of graph priors employed by various algorithms.

During the GRN inference process, graph prior edges are handled according to the confidence level attributed to them. If edges are considered certain, they are retained in the final GRN and referred to as "hard positives." Algorithms that accept graph priors with "hard positives" work in a "constructive" manner, adding missing edges to the prior graph to generate the final GRN. In contrast, some algorithms operate in a "destructive" fashion, generating the final GRN by removing edges from the graph prior. Here, the prior is treated as a list of potential interactions to be filtered, with the absence of an edge serving as evidence of no interaction (i.e., the prior contains "hard negatives"). A third approach is a "hybrid" method that allows both the addition and removal of edges from the graph prior, treating it as a "soft prior". Figure 2A provides a graphical summary of these three modes of operation.

As noted earlier, algorithms using graph-based priors offer a modular and flexible framework, allowing for customized graph priors as input. However, for effective results, it is critical to align the choice of algorithm with the nature of the graph prior (i.e., hard positives, hard negatives, or soft prior). Figure 2B summarizes the operational modes of the algorithms reviewed here. For instance, GENELink and GRGNN subsample the ground truth graph to obtain hard positive and hard negative edges (in the case of GENELink). In scPRINT, a curated set of hard positives and hard negatives is utilized to train a regression model, enabling the identification of the most effective attention heads in a transformer

model for predicting the rest of the GRN. In addition, GRGNN utilizes a soft prior on the remaining edges. Inferelator 3.0 removes certain genes from the prior network to assess performance on these genes using the ground truth later. Algorithms that operate in a "destructive" manner by removing edges from the graph prior include CellOracle, Symphony, KiMONO, and PriorPC, with PriorPC being the only one of these able to leverage a soft prior for potential interactions. Finally, algorithms capable of adding or removing edges based on a soft prior include scGLUE, iRafNet, NetREX-CF, BDgraph, D-SPIN and Graphical Lasso.

Figure 2C provides an overview of how graph priors are constructed using the types of data discussed in the section "Experimental sources of prior knowledge". Protein–protein interactions, perturbation data, DNA binding motifs, and the intrinsic feature extraction from RNA-seq are direct sources for TF-TG link priors. Alternatively, an enhancer GRN can serve as an intermediate step, consisting of TF-RE and RE-TG links, which can then be collapsed into a homogeneous TF-TG prior graph, as done in CellOracle (Kamimoto et al, 2023). Selecting functionally relevant regulatory elements involves data such as chromatin accessibility, TF ChIP-seq, evolutionary conservation across species, and enhancer activity assays. TF-RE links are typically derived from either TF binding motif enrichment or TF ChIP-seq data, while RE-TG links are estimated using genomic proximity, chromatin interaction data like Hi-C, or eQTL data.

In addition to being constructed, graph priors can also be retrieved from databases. These databases (reviewed in (Baltoumas et al, 2021)) combine inferred prior connections from various sources, including manually curated literature or predicted TF-TG links. Other databases store information critical for building priors, such as sets of known regulatory elements, like the ENCODE database (The ENCODE project consortium, 2012). GRaNIE (Kamal et al, 2023) is another example, offering a precompiled resource of potential TF binding sites for humans and mice, instead of computing motif enrichment scores directly from selected peaks. The choice of database often depends on the organism, cell type, and specific information required. For example, in iRafNet (Petralia et al, 2015), protein–protein interactions are sourced from BioGrid (Oughtred et al, 2019), DIP (Xenarios et al, 2000), and MINT (Zanzoni et al, 2002) to construct a GRN prior for *S. cerevisiae*. In KiMONO (Ogris et al, 2021), BioGrid is used to create gene-gene links, while Inferelator 3.0 (Gibbs et al, 2022) builds a GRN prior for *S. cerevisiae* based on the YEASTRACT network (Monteiro et al, 2020).

For certain well-studied models and cell types, databases of TF-TG interactions are considered "ground truth" networks due to their comprehensiveness. These "silver" or "gold" standard networks are frequently used to benchmark GRN inference methods. Algorithms that do not utilize prior knowledge compare their predicted GRN against these ground truth networks. For algorithms that do incorporate prior knowledge, part of the ground truth is typically subsampled as input, and the remaining hold-out interactions are used to evaluate the algorithm's performance (e.g., GRGNN, PriorPC, GENELink, scPRINT). One prominent example of a ground truth network is the one provided by the DREAM5 challenge (Marbach et al, 2012), which uses RegulonDB (Salgado et al, 2024) to curate a ground truth for *E. coli* (used by GRGNN and PriorPC). For *S. cerevisiae*, ChIP-seq data combined with

binding motif conservation is used to construct a ground truth (used by GRGNN and iRafNet). Other ground truth networks, such as those on mouse and human embryonic stem cells, are curated in the BEELINE benchmarking study (Pratapa et al, 2020), and used by GENELink (Chen and Liu, 2022) and (McCalla et al, 2023), and by scPRINT (Kalfon et al, 2024), which also relies on Omnipath (Türei et al, 2016) and ENCODE (The ENCODE project consortium, 2012) for additional validation. While gold standard networks are widely used, they tend to have low complexity, cover a limited range of interactions, and are often unavailable for higher eukaryotes. Consequently, benchmarking against these networks has inherent limitations (Kamal et al, 2023). For less-studied organisms, databases from closely related species may be used as a substitute, although this approach can introduce additional uncertainty (Nair, 2017).

# Benchmarking GRN inference algorithms that incorporate prior knowledge

The diversity of GRN inference algorithms and the various ways they incorporate prior knowledge provide researchers with many options. However, this variety can complicate the selection process for users and developers alike. As the field continues to grow, there is an increasing need for robust and transparent benchmarking studies that can guide the selection of methods. These studies should focus on comparing algorithms fairly, particularly those that leverage prior knowledge, as the quality and type of prior knowledge used can significantly influence the algorithm's performance. Several existing benchmarking efforts have compared GRN inference algorithms based on their ability to infer TF-TG links (Marbach et al, 2012; McCalla et al, 2023; Pratapa et al, 2020) or predict GRN topology (Stock et al, 2024). However, these studies often overlook the critical role of prior knowledge in influencing outcomes. Thus, future benchmarking studies should account for both the computational strategies and the nature of the prior knowledge used. Disentangling these factors will be essential for identifying algorithmic weaknesses and advancing the field.

To achieve a more objective comparison, we recommend standardizing the representation of prior knowledge across algorithms. For instance, the use of a homogeneous graph prior could allow for a fair evaluation of both algorithms and the quality of their prior knowledge sources (see Fig. 2D). Such an approach would enable not only the standard comparison of algorithm-prior combinations (Fig. 2D, top panel) but also two additional evaluations: a ranking of the value of different prior types and a comparable assessment of algorithmic performance using the same prior as input (Fig. 2D, bottom panel). Furthermore, this framework could enable the integration of multiple graph priors into a single input, allowing researchers to assess whether the combined use of different priors leads to improved GRN inference accuracy. For such a benchmarking study, the operation mode of the benchmark algorithms has to be taken into account, since destructive algorithms cannot be directly benchmarked against constructive algorithms. Sets of algorithms suitable for benchmarking, which share the same type of RNA expression input (e.g., scRNA-seq) and compatible operational modes (e.g., constructive or hybrid), can be identified from Table 2.

**Table 2.** Summary overview of the presented methods that leverage prior knowledge as a graph representation.

Overview of methods including a graph prior

| Algorithm | Type | Mode | Exp. Input | Graph | Code |
|---|---|---|---|---|---|
| GENELink (Chen and Liu, 2022) | Neural network (graph autoencoder) | Constr. | scRNA-Seq, TF ChIP-Seq | Homog. (binary) | Python |
| Inferelator 3.0 (Gibbs et al, 2022) | Regression (BBSR, StARS-LASSO, Multi-task Learning) | Constr. | scRNA-Seq, ATAC-Seq, Motifs, Gen. distance | Homog. (binary) | Python |
| scPRINT (preprint: (Kalfon et al, 2024)) | Neural network (Transformer) | Constr. | scRNA-Seq, Arbitr. graph | Homog. (binary) | Python |
| GRGNN (Wang et al, 2020) | Neural Network (Graph convolution) | Constr. | Microarray, Arbitr. graph | Homog. (binary) | Python |
| CellOracle (Kamimoto et al, 2023) | Regression (Bayesian Ridge or Bagging Ridge) | Destr. | scRNA-Seq, ATAC-Seq, Motifs, Gen. distance | Homog. (binary) | Python |
| Symphony (preprint: (Burdziak et al, 2019)) | PGM (Bayesian hierarchical model) | Destr. | scRNA-Seq, ATAC-Seq, Motif, Gen. distance | Homog. (binary) | Python (Jupyter) |
| PriorPC (Ghanbari et al, 2015) | PGM (Bayesian Network) | Destr. | Microarray, Arbitr. graph | Homog. (weights) | No code |
| iRafNet (Petralia et al, 2015) | Regression (Random Forest) | Hybrid | Microarray, Knockouts, PPI networks | Homog. (weights) | R |
| NetREX-CF (Wang et al, 2022) | Regression (Collaborative filtering, Network Component Analysis) | Hybrid | scRNA-Seq, Motifs, TF ChIP-Seq, Knockouts, Gen. distance | Homog. (weights) | Python (Jupyter) |
| D-SPIN (preprint: (Jiang et al, 2024)) | PGM (Max Entropy) | Hybrid | scRNA-Seq, Knockouts, Arbitr. graph | Homog. (weights) | Python +Matlab |
| KiMONO (Ogris et al, 2021) | Regression (Sparse Lasso) | Destr. | bulk RNA-Seq, Gen. distance, Protein, Methylation, SNPs, Mutation, ... | Heterog. (binary) | R |
| scGLUE (Cao and Gao, 2022) | Neural network (Graph autoencoder) | Hybrid | scRNA-Seq, ATAC-Seq, TF ChIP-Seq, Gen. distance, Hi-C, eQTLs, Methylation, ... | Heterog. (weights) | Python |
| BDgraph (Mohammadi and Wit, 2019) | PGM (Gaussian Graphical Models) | Hybrid | Microarray, Topology, Arbitr. graph | Heterog. (weights) | R |
| Graphical Lasso (Friedman et al, 2008) | PGM (Regularized Graphical Models) | Hybrid | bulk RNA-Seq, Topology, Arbitr. graph | Heterog. (weights) | R |

For each method, we also indicate the type of graph priors it utilizes (homogeneous or heterogeneous) and its operation mode (constructive, destructive, or hybrid).

**Table 3.** Summary overview of the presented methods that leverage a data prior without an intermediate graph representation of the prior

| Overview of methods including a data prior | | | |
|---|---|---|---|
| Algorithm | Type | Exp. Input | Code |
| GRaNIE (Kamal et al, 2023) | Correlation | bulk RNA-Seq, ATAC-Seq, Motifs, Histone ChIP-Seq, Gen. distance, Hi-C | R |
| FigR (Kartha et al, 2022) | Correlation | scRNA-Seq, ATAC-Seq, Motifs, Gen. distance | R |
| SCENIC+ (Bravo González-Blas et al, 2023) | Regression (Gradient boosting) | scRNA-Seq, ATAC-Seq, Motifs, Gen. distance | Python |
| Pando (Fleck et al, 2023) | Regression (linear) | scRNA-Seq, ATAC-Seq, Motifs, Sequence conservation., Gen. distance | R |
| scMEGA (Li et al, 2023b) | Correlation | scRNA-Seq, ATAC-Seq, Gen. distance | R |
| NSCGRN (Liu et al, 2022) | Mutual Information | Microarray, Topology | No code |

## Conclusion and perspectives

In this review, we provided a comprehensive overview of the current strategies employed by GRN inference algorithms that integrate scRNA-seq data with prior knowledge. We discussed the various types of prior knowledge that can be utilized, the diverse ways these priors are incorporated into algorithms, and how these approaches affect the performance of GRN inference. The accompanying tables summarize key information about the reviewed algorithms, with those incorporating graph representations of prior knowledge listed in Table 2, and those relying on other types of priors in Table 3. A detailed description of recent representative inference algorithms that leverage graph priors is in the Appendix Texts S1–S4.

By examining how different algorithms utilize diverse sources of prior knowledge and proposing a unified representation of these priors as graphs, we aim to encourage the integration of multiple resources to improve the accuracy of inferred GRNs. While scRNA-seq data serves as a rich and valuable source of biological insight, it has inherent limitations that can impact GRN inference. At the same time, the availability and quality of complementary data sources, such as curated databases and other omics data, are continually improving. Leveraging these diverse datasets alongside scRNA-seq creates a powerful opportunity to enhance GRN inference, paving the way for deeper insights into gene regulatory mechanisms, the identification of therapeutic targets, and other transformative discoveries.

One underexplored but promising aspect is the potential of graph priors to provide more detailed GRN models. Current GRN inference methods have progressed from undirected to directed networks and have begun incorporating edge annotations representing activating and inhibiting regulatory relationships. However, further refinement is possible. For instance, self-loops, which are currently under-represented in GRN models, could offer additional mechanistic insights. Similarly, current models often reduce regulatory interactions to simple OR-gate representations, neglecting cases where two TFs jointly regulate a target gene, which would be better represented by an AND-gate. These complexities, which remain unaccounted for in both TF-TG networks and enhancer-based GRNs, could be more effectively modeled using hypergraphs. Prior knowledge, for example from perturbation experiments, could help distinguish between co-regulation of a target by two TFs and independent regulation of the target by each TF. Advances in machine learning methods for hypergraph-based inference, combined with comprehensive

perturbation datasets-such as combinatorial knockout assays-will be essential to achieve this level of detail.

Building on our analysis and classification of GRN inference algorithms, we proposed a benchmarking strategy specifically designed for algorithms that incorporate prior knowledge (see the section "Benchmarking GRN inference algorithms that incorporate prior knowledge"). This approach aims to disentangle the individual contributions of the type of prior knowledge and the algorithm itself to overall performance. Such a benchmarking study is currently lacking but is needed to address critical gaps in the field. Another essential element of future benchmarking strategies is the emphasis on openness, reproducibility, and transparency, which are vital for driving progress in GRN inference research. Initiatives like the Open Problems framework (Luecken et al, 2024) embody these principles, offering a strong foundation and setting an important precedent for future efforts. By adhering to these values, benchmarking studies can deliver robust, comparable, and broadly applicable evaluations of GRN inference methods, ultimately advancing the field.

Looking forward, we anticipate that future GRN inference methods will draw upon and combine ideas from the approaches we reviewed, addressing their current limitations while leveraging increasingly standardized benchmarking frameworks. As high-quality single-cell datasets become more available, especially with the rise of pre-trained foundation models (Cui et al, 2024; Theodoris et al, 2023), the inclusion of RNA-independent priors will further enhance our understanding of GRNs.

## Peer review information

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

## Acknowledgements

MS and CL were supported by the Helmholtz Association under the joint research school "Munich School for Data Science - MUDS". MS was supported by a Joachim Herz Stiftung Add-on Fellowship for Interdisciplinary Life Science. This work was supported by the Helmholtz Association (AS, EH) and the Deutsche Forschungsgemeinschaft (SC280/2-1 to AS, HO 6864/2-1 and CRC1064 Chromatin Dynamics Project-ID 213249687 to EH). This project has been made possible in part by the Chan Zuckerberg Foundation (2019-202666, 2021-237882 to MH) and was supported by the DZHK partner site project 81Z0600106 (MH).

## Author contributions

**Marco Stock**: Conceptualization; Formal analysis; Investigation; Visualization; Methodology; Writing—original draft; Writing—review and editing. **Corinna Losert**: Formal analysis; Investigation; Methodology; Writing—original draft; Writing—review and editing. **Matteo Zambon**: Formal analysis; Investigation; Methodology; Writing—original draft. **Niclas Popp**: Formal analysis; Investigation; Methodology; Writing—original draft. **Gabriele Lubatti**: Formal analysis; Investigation; Methodology; Writing—original draft. **Eva Hörmanseder**: Supervision; Writing—review and editing. **Matthias Heinig**: Supervision; Writing—review and editing. **Antonio Scialdone**: Conceptualization; Supervision; Funding acquisition; Project administration; Writing—review and editing.

## Disclosure and competing interests statement

The authors declare no competing interests.

