## [Peer Review File · Molecular Systems Biology]

Leveraging prior knowledge to infer Gene Regulatory Networks from single-cell RNA-sequencing data

Marco Stock, Corinna Losert, Matteo Zambon, Niclas Popp, Gabriele Lubatti, Eva Hörmanseder, Matthias Heinig, and Antonio Scialdone

Corresponding author(s): Antonio Scialdone (antonio.scialdone@helmholtz-munich.de)

Review Timeline:

Submission Date:	10th Oct 24
Editorial Decision:	8th Nov 24
Revision Received:	17th Dec 24
Editorial Decision:	23rd Jan 25
Revision Received:	29th Jan 25
Accepted:	30th Jan 25

Editor: Poonam Bheda

Transaction Report:

8th Nov 2024

Manuscript Number: MSB-2024-12676

Title: Leveraging prior knowledge to infer Gene Regulatory Networks from single-cell RNA-sequencing data

Dear Dr Scialdone,

Thank you for your submission to Molecular Systems Biology.

Please find below the two sets of comments I have now received regarding your piece. As you will see, the referees are positive regarding the topic, but they do raise several important concerns that we would like you to address in a revision. In particular Reviewer 1 finds the motivation and unique angle of the review somewhat unclear and Reviewer 2 is unsure of the target audience, requesting expansion of introductions. Editorially we also generally felt that some aspects of the review could be further expanded for more clarity to the reader.

Please also ensure that all reviewer concerns are addressed. We may send your Review back to the reviewers to ensure that the motivation and angle of the review and target audience are sufficiently clarified.

- 1) a .docx (or LaTeX) formatted version of the manuscript text (including Figure legends and tables)
- 2) Separate figure files. Please remove all figures from main manuscript file and leave only the figure legends placed after the references.

We will send the article to our graphic artist who will edit/re-draw the figures for style and clarity. Therefore, please ensure the information shown is scientifically accurate and upload the file as a PDF (or SVG, or EPS), PowerPoint or Keynote in which the labels and objects are still editable. For figures created using Adobe Illustrator, please send the Illustrator (.ai) file. You can also send these to me by email (or share via link for files that are too big) so that we can already send these to the graphic designer to prevent delay in publishing your manuscript.

- 3) Up to 5 keywords.

4) Please reorder the manuscript as follows: keywords should be below abstract; acknowledgements with funding info and disclosure and competing interests statement should be before references, and main figure legends and tables below them; corresponding author's email address should be placed below affiliations on the title page

5) Please rename "Conflict of Interests" to "Disclosure and competing interests statement". We updated our journal's competing interests policy in January 2022 and request authors to consider both actual and perceived competing interests. Please review the policy <https://www.embopress.org/competing-interests> and update your competing interests if necessary.

6) Author contributions: Please remove it from the manuscript and specify author contributions in our submission system. CRediT has replaced the traditional author contributions section because it offers a systematic machine-readable author contributions format that allows for more effective research assessment. You are encouraged to use the free text boxes beneath each contributing author's name to add specific details on the author's contribution. More information is available in our guide to authors:

<https://www.embopress.org/page/journal/17574684/authorguide#authorshipguidelines>

7) References: Please correct the reference citation in the reference list. References should be alphabetical, not numerical, and when there are more than 10 authors on a paper, only the first 10 should be listed, followed by "et al.". Please check "Author Guidelines" for more information.

<https://www.embopress.org/page/journal/17574684/authorguide#referencesformat>

8) As part of the EMBO Publications transparent editorial process initiative (see our policy here:

https://www.embopress.org/transparent-process#Review_Process), Molecular Systems Biology will publish online a Peer Review File (PRF) to accompany accepted manuscripts. This file will be published in conjunction with your paper and will include the anonymous referee reports, your point-by-point response and all pertinent correspondence relating to the manuscript. Let us know whether you agree with the publication of the PRF and as here, if you want to remove or not any figures from it prior to publication. Please note that the Authors checklist will be published at the end of the PRF.

9) Please provide a point-by-point letter INCLUDING my comments as well as the reviewer's reports and your detailed

responses (as Word file).

If you have any questions, please don't hesitate to ask. I look forward to seeing the revised manuscript.

Yours sincerely,

Poonam Bheda

Poonam Bheda, PhD
Scientific Editor
Molecular Systems Biology

Reviewer #1:

In the review manuscript "Leveraging prior knowledge to infer Gene Regulatory Networks from single-cell RNA-sequencing data", the authors summarize the recent development on inferring Gene Regulatory Networks (GRNs), especially on how they utilize prior knowledge. The authors first introduce the various types of GRNs and inference methods, which could be divided into several categories according to the nature of the algorithms. Then, authors describe how prior knowledge is generated and how it could be leveraged to enhance the performance of GRN inference methods. Finally, the authors raise several limitations on current GRN inference methods and suggestions on how prior knowledge can be adopted in benchmarking the performance of these methods. While there are several novel and meaningful elements, overall several limitations in the current version may prevent this review from being published.

Major issues:

1. It seems the motivation of this review is unclear: though the authors briefly introduce the focus of recent reviews, the limitations of them are not highlighted and the knowledge gap is not described. The authors need to point out the limitations explicitly, which will help to understand the motivation of this review.
2. In the Introduction section, the authors cite several benchmark studies and claim that they are still limited. It would be better to specify their limitations. Related to this, authors' suggestions on the benchmark studies in Conclusion and Perspectives could be moved to an individual section, with complete descriptions on the limitations, suggestions and essential citations.
3. In Section 2.2, 'Types of Inference Algorithms,' the authors categorize various methods within four main groups but provide limited detail on the specific characteristics and distinctions of methods within each category. Expanding on the unique aspects and differences among these methods could enhance clarity and provide a more comprehensive understanding of each category.
4. When describing the source of prior knowledge, the authors regarded the transcription factors as a standard input and didn't elaborate much on them. How is the list created and justified? What's the difference among the TF lists from diverse databases? It would be better to add transcription factors as a subsection in 3.1 "Experimental sources of prior knowledge".
5. It would be better to briefly describe how each type of experimental source is generated from experiments.
6. It would be better to illustrate the workflow of representative prior-based GRN inference methods using figures or pseudo-codes.
7. In section 3.3, the authors need to include necessary details to justify why graph-based priors offer greater versatility in integrating diverse types of prior knowledge and data-based priors cannot, to improve the clarity.
8. In the fourth paragraph of Conclusion and Perspectives, the authors list several aspects where GRN inference algorithms could be improved. However, the relationship between these aspects and the utilization of prior knowledge is unclear in the review.
9. The organization and structure of the whole manuscript could be further improved. For example, in Section 3.1, the technology that offers prior knowledge, the databases that use the technology, the inference algorithms, and the applications are usually intertwined, without clear logic. Besides, the paragraphing of Section 3.4 seems problematic.
10. It is suggested that the language of the manuscript be checked thoroughly to avoid redundancy, repetition and pronoun usage. For example, "additional information or data beyond gene expression data" and "common information available in addition to the gene expression data" in Section 3 seems redundant. Some expressions are rather vague, like the second paragraph of Section 3.1, as they do not convey any new information.

Minor issues:

1. Some sentences are not complete and have grammar errors. For example, "First, we categorize the available sources of prior knowledge and the strategies algorithms use to incorporate them into the inference process." In section 3, "different types of

prior..." should be "different types of priors".

2. Some of the abbreviations of terminology are repeatedly introduced in the manuscript. Besides, it would increase the readability if all the abbreviations are summarized in a separate table.
3. In Section 3, authors should specify the name of the databases instead of merely citing them (e.g., Ref. 53).
4. What does "following" refer to in "The following single-cell mouse neuron network inference was done using the TRANSFAC database ..." (Section 3.1.2).
5. In the table of Fig. 2b, there is a double-border gap between NetREX-CF and KIMONO. It's unclear what's the difference between the two groups separated by the gap.

Reviewer #2:

Inference of gene regulatory networks is one of the hottest topics currently in the field of synthetic biology, biophysics and machine learning. The huge amount and variety of available data continuously calls for more refined techniques aimed at combining information from different experimental sources. The past decade(s) has seen a high proliferation of methods developed to this aim. Yet, a comprehensive review discussing general issues in inference of gene regulatory networks, available datasets and listing the advantages and limitations of different data integration as well as the different most used and available methods is missing. In this review the authors provide the readers with an in-depth discussion of each of these points. In particular, they focus on how leveraging prior knowledge to infer gene regulatory networks in the context of the use of single-cell RNA-seq data. After a brief introduction on the general questions, gene regulatory networks and the importance of inferring them the authors first list the different types of algorithms currently available and the principles on which they are based. They continue by discussing the importance of leveraging prior knowledge and list all different types of available data (e.g., ATAC-seq, TF-Motif Enrichment, ChIP-Seq, etc.) as well as algorithms using each of (or combinations of) them in order to give a comprehensive framework. They conclude by discussing current limitations and future advances.

I definitely like this review, which is really well written, clear and exhaustive. Also, I think that the field currently needs such a comprehensive review. However, I have a few major and minor comments which I list hereby.

Major comments:

- It is not clear to me what is the target reader the authors are aiming to. For each paragraph, the authors first start with a short yet general introduction and then quickly switch to a very detailed description of datasets and algorithms. Personally, since the review is really well written and exhaustive, I would try if possible to give a bit more detail in the still short intros. For example, by defining gene regulatory networks and their structure and the exact structure of RNA-seq data, their features, advantages and limitations, and citing papers in this context.
- Among the algorithms, neither of those using Max Entropy Principles are cited. I would give some space to this type of methods as well.
- Upon discussing pre-processing of RNA-seq data, why not mentioning dimensional reduction by methods such as PCA, t-SNE and UMAP? Also, I believe that it is important to mention methods aimed at batch corrections.
- For each paragraph concerning the different types of datasets I would introduce first what type of question integrating this type of dataset with RNA-seq would answer.

Minor comments:

- In paragraph 2.2 the authors rightly say that Gaussian distributions may not hold for single-cell RNA-seq data which I totally agree with of course. Yet, it would be good to say why and what kind of distributions are normally over-represented (I really liked the discussion from this paper doi: 10.1093/bioinformatics/btz177 .
- In paragraph 3.1.4: don't -> do not.

We sincerely thank both Reviewers for their thorough evaluation of our manuscript and their valuable suggestions. Below, we provide a detailed point-by-point response to all their comments, with our replies presented in blue italics for clarity. Major revisions made to the manuscript are highlighted in red in the updated version, with the corresponding line numbers referenced in this document.

Reviewer #1

In the review manuscript "Leveraging prior knowledge to infer Gene Regulatory Networks from single-cell RNA-sequencing data", the authors summarize the recent development on inferring Gene Regulatory Networks (GRNs), especially on how they utilize prior knowledge. The authors first introduce the various types of GRNs and inference methods, which could be divided into several categories according to the nature of the algorithms. Then, authors describe how prior knowledge is generated and how it could be leveraged to enhance the performance of GRN inference methods. Finally, the authors raise several limitations on current GRN inference methods and suggestions on how prior knowledge can be adopted in benchmarking the performance of these methods. While there are several novel and meaningful elements, overall several limitations in the current version may prevent this review from being published.

We thank this Reviewer for acknowledging the "several novel and meaningful elements" in our manuscript and for all the important and constructive criticisms, which we have addressed in the revised text, as explained below.

1. It seems the motivation of this review is unclear: though the authors briefly introduce the focus of recent reviews, the limitations of them are not highlighted and the knowledge gap is not described. The authors need to point out the limitations explicitly, which will help to understand the motivation of this review.

Our review addresses two significant gaps in the current literature on GRN inference. First, while most existing reviews focus heavily on using single-cell multiomics data (especially ATAC-seq)—a pivotal approach to GRN inference—this is only one of many methodologies under development. The incorporation of prior knowledge into GRN inference is a broader and more diverse topic. In this review, we explore a wider range of algorithms and data types, providing a much more comprehensive perspective than previous reviews.

Second, the rapid development of new algorithms has made it increasingly challenging to discern their differences and evaluate their performance effectively. To address this, we analyze the various computational approaches, classifying algorithms based on the types of priors they utilize ("data" and "graph" priors) and their operational modes ("constructive", "destructive," or "hybrid"). This classification has informed our proposal of a novel benchmarking strategy, which we detail in a new standalone section (section 5) of the revised manuscript.

To address this comment, we have highlighted the motivation for this review with a new paragraph in the Introduction section (Lines 48-61).

2. In the Introduction section, the authors cite several benchmark studies and claim that they are still limited. It would be better to specify their limitations. Related to this, authors' suggestions on the benchmark studies in Conclusion and Perspectives could be moved to an individual section, with complete descriptions on the limitations, suggestions and essential citations.

We recognize the importance of providing readers with more detailed information about existing benchmarking studies and the limitations of the GRN algorithms evaluated. To address this, we have added a paragraph in the Introduction (Lines 32-37) that elaborates more on the main limitations of GRN inference algorithms identified in current benchmarking studies and the lack of an objective and unbiased benchmarking framework for evaluating GRN algorithms using prior knowledge (Lines 54-61; Lines 686-692).

Additionally, we have added a new section titled "Benchmarking GRN Inference algorithms that incorporate prior knowledge" (section 5) that includes our recommendations for improving benchmark studies with citations and discussion over existing benchmarking studies.

3. In Section 2.2, 'Types of Inference Algorithms,' the authors categorize various methods within four main groups but provide limited detail on the specific characteristics and distinctions of methods within each category. Expanding on the unique aspects and differences among these methods could enhance clarity and provide a more comprehensive understanding of each category.

We agree with this comment and have revised Section 2.2 to include more detailed explanations of the specific features of each algorithm discussed (Lines 141-144; 146-151; 153-168; 173-180; 182-187)

4. When describing the source of prior knowledge, the authors regarded the transcription factors as a standard input and didn't elaborate much on them. How is the list created and justified? What's the difference among the TF lists from diverse databases? It would be better to add transcription factors as a subsection in 3.1 "Experimental sources of prior knowledge".

We fully agree that this is very useful information to add, and following the Reviewer's suggestion, we have added a new subsection (subsection 3.1, "Transcription Factor Databases") where we include the various sources of TF lists and their differences.

5. It would be better to briefly describe how each type of experimental source is generated from experiments.

There is now a short description of each experimental technique we refer to and the data that it generates (section 3.2, Lines 282-286; 314-321; 347-353; 384-393; 435-437; 446-453; 502-509).

6. It would be better to illustrate the workflow of representative prior-based GRN inference methods using figures or pseudo-codes.

In addition to including now a short description of all the algorithms we mention (see point 3 above), we illustrate in more detail a selection of algorithms and their workflows in a new supplementary section (8.2, "Description of representative algorithms"). This selection includes the most recent (> 2022) and cited graph-prior algorithms, covering all three "operation modes" we discuss.

7. In section 3.3, the authors need to include necessary details to justify why graph-based priors offer greater versatility in integrating diverse types of prior knowledge and data-based priors cannot, to improve the clarity.

The reason is that employing a unified graph representation of the prior as input enables the seamless integration of priors derived from diverse experimental sources without requiring modifications to the algorithm. Additionally, this approach facilitates the disentanglement of performance improvements due to the prior from those due to the inference algorithm itself in benchmarking studies. We have added this more detailed justification to section 3.4 (Lines 593-598).

8. In the fourth paragraph of Conclusion and Perspectives, the authors list several aspects where GRN inference algorithms could be improved. However, the relationship between these aspects and the utilization of prior knowledge is unclear in the review.

Thanks for highlighting this point. This paragraph was intended to outline key open challenges in GRN inference algorithms, both with and without the use of prior knowledge. To better align the discussion with the focus of the review, in the revised manuscript, we have provided additional context, explicitly addressing the connection between these challenges and the role of prior knowledge (Lines 734-736).

9. The organization and structure of the whole manuscript could be further improved. For example, in Section 3.1, the technology that offers prior knowledge, the databases that use the technology, the inference algorithms, and the applications are usually intertwined, without clear logic. Besides, the paragraphing of Section 3.4 seems problematic.

We agree that the organization and structure could be improved and have revised the manuscript accordingly. Specifically, we have reorganized the subsections to follow a clearer and more consistent logical flow (section 3.2, Lines 271-275). To aid readers, we have included a brief introductory paragraph at the beginning of each section, outlining its structure and purpose. Additionally, we have addressed the paragraphing issues in Section 3.4 by removing unnecessary breaks to enhance readability.

10. It is suggested that the language of the manuscript be checked thoroughly to avoid redundancy, repetition and pronoun usage. For example, "additional information or data beyond gene expression data" and "common information available in addition to the gene expression data" in Section 3 seems redundant. Some expressions are rather vague, like the second paragraph of Section 3.1, as they do not convey any new information.

Thank you for pointing this out. We have thoroughly proofread the manuscript to address these issues and convey clearer and more concise information.

Minor issues:

1. Some sentences are not complete and have grammar errors. For example, "First, we categorize the available sources of prior knowledge and the strategies algorithms use to incorporate them into the inference process." In section 3, "different types of prior..." should be "different types of priors".
2. Some of the abbreviations of terminology are repeatedly introduced in the manuscript. Besides, it would increase the readability if all the abbreviations are summarized in a separate table.
3. In Section 3, authors should specify the name of the databases instead of merely citing them (e.g., Ref. 53).
4. What does "following" refer to in "The following single-cell mouse neuron network inference was done using the TRANSFAC database ..." (Section 3.1.2).
5. In the table of Fig. 2b, there is a double-border gap between NetREX-CF and KIMONO. It's unclear what's the difference between the two groups separated by the gap.

We have addressed all the minor points the Reviewer pointed out, including the addition of a list of abbreviations (section 8.1).

Reviewer #2

Inference of gene regulatory networks is one of the hottest topics currently in the field of synthetic biology, biophysics and machine learning. The huge amount and variety of available data continuously calls for more refined techniques aimed at combining information from different experimental sources. The past decade(s) has seen a high proliferation of methods developed to this aim. Yet, a comprehensive review discussing general issues in inference of gene regulatory networks, available datasets and listing the advantages and limitations of different data integration as well as the different most used and available methods is missing. In this review the authors provide the readers with an in-depth discussion of each of these points. In particular, they focus on how leveraging prior knowledge to infer gene regulatory networks in the context of the use of single-cell RNA-seq data. After a brief introduction on the general questions, gene regulatory networks and the importance of inferring them the authors first list the different types of algorithms currently available and the principles on which they are based. They continue by discussing the importance of leveraging prior knowledge and list all different types of available data (e.g., ATAC-seq, TF-Motif Enrichment, ChIP-Seq, etc.) as well as algorithms using each of (or combinations of) them in order to give a comprehensive framework. They conclude by discussing current limitations and future advances.

I definitely like this review, which is really well written, clear and exhaustive. Also, I think that the field currently needs such a comprehensive review. However, I have a few major and minor comments which I list hereby.

We thank this Reviewer for all the positive comments on our manuscript. Below, we provide a detailed answer to the points raised, alongside a list of changes made in the revised text.

1. It is not clear to me what is the target reader the authors are aiming to. For each paragraph, the authors first start with a short yet general introduction and then quickly switch to a very detailed description of datasets and algorithms. Personally, since the review is really well written and exhaustive, I would try if possible to give a bit more detail in the still short intros. For example, by defining gene regulatory networks and their structure and the exact structure of RNA-seq data, their features, advantages and limitations, and citing papers in this context.

Thanks for all these suggestions. We implemented all of them, and we think that they have further improved our manuscript's clarity.

First, we have stated more clearly in the Introduction that our intended readership includes both researchers who apply GRN inference algorithms in their studies and those interested in developing novel computational methodologies (Lines 71-79). This dual focus reflects the scope and balance of our review, which covers practical application insights with a description of the algorithmic strategies. Furthermore, we have significantly extended the introductory paragraphs adding details on the topics suggested by the Reviewers. In particular, we added a new paragraph discussing more in detail the opportunities and challenges that single-cell RNA-seq data entail to make the review more accessible to a broader audience (Lines 207-224).

2. Among the algorithms, neither of those using Max Entropy Principles are cited. I would give some space to this type of methods as well.

We have now added the “D-SPIN” algorithm, a recent preprint using spin networks, which are a type of maximum entropy model. The algorithm fits nicely in the scope of the review, since it is capable of leveraging data from perturbation experiments and can also incorporate weighted transcription factor to target genes graph-priors (Lines 167-168; 373-374; 384-385; 588-592; 605-611; 639-641).

3. Upon discussing pre-processing of RNA-seq data, why not mentioning dimensional reduction by methods such as PCA, t-SNE and UMAP? Also, I believe that it is important to mention methods aimed at batch corrections.

We agree with this comment and have mentioned these important topics in a new paragraph at the end of section 2.2 (Lines 207-224).

4. For each paragraph concerning the different types of datasets I would introduce first what type of question integrating this type of dataset with RNA-seq would answer.

Thanks, we made sure the goal of each dataset integration is mentioned in the sections discussing the different types of data (Lines 271-271; 312-313; 513-517).

Minor comments:

- In paragraph 2.2 the authors rightly say that Gaussian distributions may not hold for single-cell RNA-seq data which I totally agree with of course. Yet, it would be good to say why and what kind of distributions are normally over-represented (I really liked the discussion from this paper doi: 10.1093/bioinformatics/btz177 .
- In paragraph 3.1.4: don't -> do not.

Thanks, we have implemented all the suggested changes.

23rd Jan 2025

Manuscript Number: MSB-2024-12676R

Title: Leveraging prior knowledge to infer Gene Regulatory Networks from single-cell RNA-sequencing data

Dear Antonio,

Thank you for the submission of your revised manuscript to Molecular Systems Biology. We have now received the enclosed reports from the referees that were asked to re-assess it. As you will see the reviewers are now globally supportive and I am pleased to inform you that we will be able to accept your review pending the following final amendments:

- 1) All tables should be placed below figure legends - please move Table 1 so that it is below the figure legends.
- 2) We do not allow 'Supplementary material' sections to be included in the main manuscript. Please organize this into an Appendix. We do not have an abbreviations section, so please explain these directly in the main text at the time they are first introduced. The remaining 8.2.x text should be renamed as Appendix Text S1, etc and compiled into a single Appendix file with a title page and the title "Appendix for (Title)" as well as a table of contents including page numbers. The Appendix should be uploaded as a separate document in PDF format. Please also ensure that the callouts to the Appendix Text(s) are also updated in the main manuscript.
- 3) In discussing the figures for your review, we have decided not to have them redrawn by our graphic artist, as they are already of good quality. We did however notice that the fonts are sometimes inconsistent in size and bolding and would suggest that you fix these small differences if they were not intended and upload publication-ready figures (e.g. Figure 1, panel C, purple box has bolded and smaller text than the other boxes).
- 4) Please let us know as to whether you agree to the publication of a Peer Review File to accompany your published review, including the anonymous referee reports, your point-by-point response and all pertinent correspondence relating to the manuscript. Please also let us know if you want to remove or not any figures from it prior to publication.
- 5) Please provide a point-by-point letter INCLUDING my comments and your responses.

I look forward to reading a new revised version of your manuscript as soon as possible.

Yours sincerely,

Poonam Bheda, PhD
Scientific Editor
Molecular Systems Biology

Reviewer #1:

The revision has addressed all my comments.

Reviewer #2:

The authors have addressed all my concerns, major and minor, with great attention and precision. I have no further comments.

All editorial and formatting issues were resolved by the authors.

30th Jan 2025

Manuscript number: MSB-2024-12676RR

Title: Leveraging prior knowledge to infer Gene Regulatory Networks from single-cell RNA-sequencing data

Dear Dr Scialdone,

Congratulations on an excellent review, I am pleased to inform you that your manuscript has been accepted for publication in Molecular Systems Biology. Thank you for your comprehensive response to referee concerns and our editorial formatting requests. It has been a pleasure to work with you to get this to the acceptance stage.

Your manuscript will be processed for publication by EMBO Press. It will be copy edited and you will receive page proofs prior to publication.

Should you experience any difficulty, please email publishing@embo.org.

If you have any other questions, please do not hesitate to contact the Editorial Office. Thank you for your contribution to Molecular Systems Biology.

Yours sincerely,

Poonam Bheda, PhD
Scientific Editor
Molecular Systems Biology
